# High-Grade Flake Graphite Deposits in Metamorphic Schist Belt, Central Finland—Mineralogy and Beneficiation of Graphite for Lithium-Ion Battery Applications

**Thair Al-Ani [1],\*, Seppo Leinonen [2], Timo Ahtola [1] and Dandara Salvador [3]**



[1]    Geological Survey of Finland (GTK), Vuorimiehentie 5, 02151 Espoo, Finland; timo.ahtola@gtk.fi
[2]    Geological Survey of Finland (GTK), Neulaniemientie 5, 70211 Kuopio, Finland; seppo.leinonen@gtk.fi
[3]    GTK-Mintec Laboratories, Tutkijankatu 1, 83500 Outokumpu, Finland; dandara.salvador@gtk.fi
[*]    Correspondence: thair.alani@gtk.fi

**Abstract:** More than 40 m length of drill cores were collected from four boreholes drilled by Geological Survey of Finland (GTK) and Outokumpu Oy in high-grade metamorphic rocks of Rautalampi and Käypysuo, Central Finland. The hosted rocks of the graphite mineralization were mica–quartz schist and biotite gneiss. The graphite-bearing rocks and final concentrated graphite powder were studied with petrographic microscope, scanning electron microscope (SEM-EDS), Raman spectroscopy, and X-ray analysis (XRD and XRF). A majority of the studied graphite had a distinctly flakey (0.2–1 mm in length) or platy morphology with a well-ordered crystal lattice. Beneficiation studies were performed to produce high-purity graphite concentrate, where rod milling and froth flotation produced a final concentrate of 90% fixed carbon with recoveries between 67% and 83%. Particle size reduction was tested by a ball and an attritor mill. Graphite purification by alkaline roasting process with 35% NaOH at 250 °C and leached by 10% $H_2SO_4$ solution at room temperature could reach the graphite purity level of 99.4%. Our analysis suggested that purifying by multistage flotation processes, followed by alkaline roasting and acid leaching, is a considerable example to obtain high-grade graphite required for lithium-ion battery production.

**Keywords:** flake graphite; flotation; purification; acid leaching; alkaline roasting; battery minerals

## 1. Introduction

Graphite occurs naturally in the Earth's crust in schist and gneiss metamorphic rocks. The graphite can have a microcrystalline structure and flaky morphology, displaying a polymorphic phase with hexagonal and rhombohedra layers. Based on its structure properties, graphite is applied in a variety of technological applications including lithium-ion batteries, fuel cells, two-dimensional grapheme, electronics, fiber optics, electrical vehicles, and so forth.

Graphite is an essential component of commercial lithium-ion batteries in the near-to-mid-term future. The vast majority of lithium-ion (Li-ion) batteries use graphite powder as an anode material. Graphite anodes meet the voltage requirements of most Li-ion cathodes, as they are relatively affordable, extremely light, porous, and durable. Recently, natural graphite has been considered as a promising anode material due to its high reversible capacity, cycle stability, higher purity, and more suitable particle size distribution [1,2].

Particle or flake size, carbon content, and grade of graphite products are important in commercial interest of the batteries industry. Graphite electrodes (anodes and cathodes) can only be produced from natural graphite ores by several beneficiation processes, which include repeated crushing, milling,

and flotation to separate the graphite flakes from their ore body. Ultra-high-purity (>99.95% C) with fine particle size ranging from 10 to 30 μm of battery grade could be achieved by further purification with alkali roasting pretreatment and acid leaching process [3]. In practical terms, graphite is one of the easiest minerals to segregate into a rough concentrate, but one of the most difficult to refine into a commercially useful product. To overcome this problem, some new grinding processes, such as vibration [4] and stirred milling [5], can be used to reduce the large flakes promptly after each flotation process while keeping the crystallinity of graphite.

Graphite-bearing rocks are rarely found as outcrops due to their softness and low weathering resistance. It has been found that geophysical investigations, especially electromagnetic, are a very effective method for locating unexposed graphite deposits in prospecting of graphite deposits. The Rautalampi and Käpysuo areas are located in the Savo Schist Belt that comprises metasedimentary and volcanic sequences, which are related to the rifting of the Archean Karelian Craton [6,7].

The rocks hosting significant graphite mineralization occurrences in Rautalampi and Käpysuo are quartz–mica schist and feldspathic biotite gneiss (Figure 1). These rocks are associated with garnet–sillimanite gneisses and garnet ± cordierite ± orthoamphibole/orthopyroxene (GCO) rocks and gneissic tonalite [8]. According to previous studies, the graphite-bearing rocks, which include volcanic rocks and metasediments from Pyhäsalmi to Rautalampi, have been dated at 1922 ± 12 Ma [9–11]. These rock types are regarded as the basement of the overlying volcano sedimentary supracrustal sequence [12].

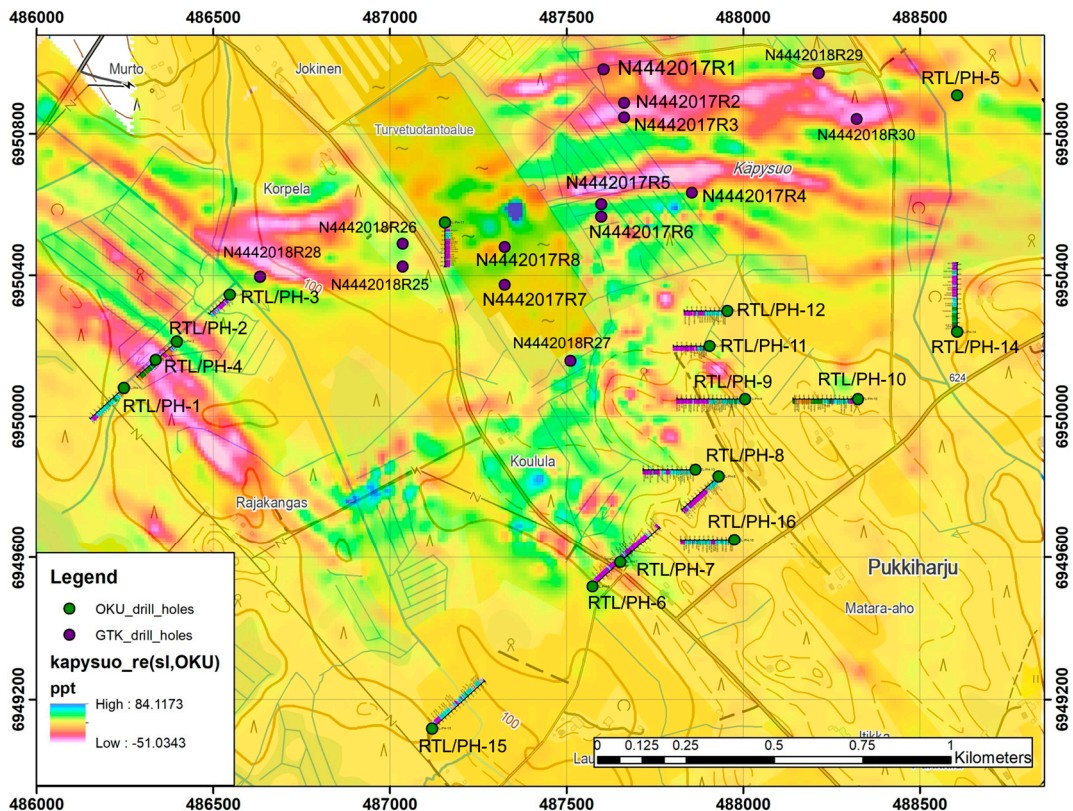

**Figure 1.** Magnetic map of the studied area. Finland–DigiKP 2018 and EM GEM-2 profiles (in-phase 1475 Hz), boreholes, and electrical resistivity tomography (ERT) profiles. Background: electromagnetic slingram in-phase component, LiDAR data (© National Land Survey of Finland).

The flake graphite deposits occur in different parts of the Proterozoic supracrustal continent with various grade graphite ores and different types of ore deposits [13,14]. According to the USA and European Union, graphite is considered to be a critical material for industry and national security. The refractories industry is the leading consumer of crystalline flake graphite, where the

graphite, having an excellent temperature resistance and stability, is used in furnace lining applications. Crystalline flake graphite can be divided into two main grades: coarse flakes (≥150–850 μm in diameter) and fine flakes (≥45–150 μm in diameter), which may be further subdivided into fractions ≥100–150 μm, ≥75–100 μm, and ≥45 μm [15–17]. Flakes in the size range of 250–1000 μm in diameter demand the highest price [18].

In this paper we describe flake graphite occurrences in Rautalampi and Käpysuo as the highest potential of flake graphite in Finland due to the quality of the hosted bedrocks and the suitable metamorphic grade. The study also describes the separation and purification techniques for producing high-quality graphite with very low concentration of impurities for lithium-ion battery anodes application requirements.

## 2. Materials and Methods

### 2.1. Sample

In the studied area, two types of rocks contain a significant amount of graphite, which include graphite flakes occurring in strongly foliated black schist, and disseminated graphite flakes within a banded gneiss. Bimodal flake graphite populations have been seen in thin section samples from Rautalampi and Käpysuo graphite deposits. The larger flakes are in excess of 1 mm long, while the small flakes are about 0.1 mm in length. The graphite is intimately associated with varying amounts of biotite, chlorite, quartz, and feldspar. Minor minerals are chlorite, pyrite, titanite, and hornblende, and accessory phases are apatite, pyroxene, zircon, and opaque.

The feed samples used in this study were obtained from a more than 40 m length of drill cores in Rautalampi and Käpysuo graphite ore for the grinding and flotation tests. Composite drill-core samples were divided into 700 g subsamples for flotation testing in GTK Mintec laboratory (Table S1). Many analytical instruments were used to identify the feed samples before and after the experiment. XRF and ICP-Ms analyses were used for chemical analysis, X-ray diffraction (XRD) was used for mineral composition, and SEM was used for morphological and elementary analyses.

### 2.2. Chemical and Mineralogical Analysis

The samples were analyzed at Eurofins Labtium Oy, Finland, using XRF and ICP-MS. The noncarbonate carbon was analyzed by the pyrolysis method (Eltra analyzer). The whole rock composition of selected samples are presented in Table S2.

The mineralogical characterization and textural relations of the minerals in the Rautalampi Käpysuo deposit were carried out on both graphite-bearing rock samples and final graphite concentrates. XRD analysis of ground samples were subjected to a Burker D8 Discover A25 X-ray diffractometer, hosted in GTK's mineralogical laboratory. The equipment parameter of copper tube source (40 kV and 40 mA), Cu Kα (Cu $K_{\alpha1}$ = 1.5406 Å; Cu $K_{\alpha2}$ = 1.5444 Å; Cu $K_\alpha$ average = 1.5418 Å; and Cu $K_\beta$ contamination = 1.3922 Å). The XRD patterns were recorded in the 2°–70° angular interval in continuous measurement mode of 0.01 2 θ/s angular velocity, with scan rate of 2°/min and count time of 0.5 s/step (Figure 2). Quantitative analysis was performed by using Bruker EVA software and ICDD (International Center for Diffraction Data), Powder Diffraction File PDF-4 Minerals 2018 database that contains only naturally occurring inorganic crystalline phases.

The scanning electron microscope (SEM-EDS), JEOL5900 LV with X-ray (EDX) detector at the GTK mineral laboratory, Espoo, Finland, was used to examine 16 thin sections made from drill cores and the graphite flakes powder extracted by flotation, leaching, and roasting processes. Backscattered imaging (BSI) was used to characterize the mineral morphology and quantity proportion of primary minerals, while the energy dispersive X-ray analysis (EDX) was used for elemental analysis. The goal was to determine if the graphite-bearing rocks differ from each other in flake size, characteristics of the graphite flakes, and their associated impurities.

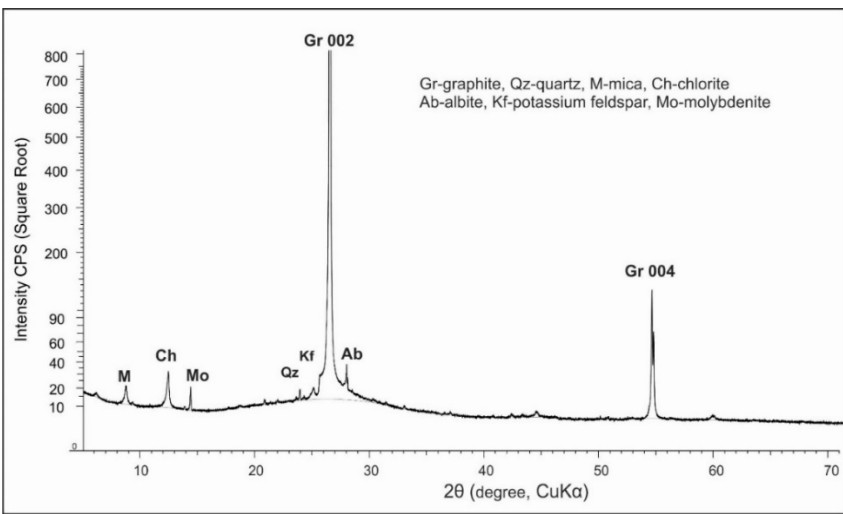

**Figure 2.** X-ray Powder Diffraction (XRD) pattern of the sample N4442017_R7.

*2.3. Raman Spectroscopy*

The Raman spectra of each sample were recorded using a Renishaw inVia Confocal Raman spectrometer equipped with a Leica DMLM microscope connected to a Leica camera, objectives 5×, 20×, 50×, and 100×, at the GTK Mintec mineral processing laboratory, Outokumpu, Finland. The measurements were made by using an argon ion laser (785/532) with the extraction wave length of 532 nm at room temperature with a laser power of 5 mW and spectrum resolution of approximately 2 cm$^{-1}$. The spectrum was calibrated against silicon water standard (520.6 cm$^{-1}$).

The analyses were on 16 thin sections and 6 polished sections that were prepared from separated graphite flakes. The most intense features of Raman spectra of graphite are visible at the first-order region, where the so-called G band and D bands are [19–21]. The G band is characteristic of in-plane vibrational mode involving sp2-hybridized carbon atoms which comprise the grapheme sheets in the graphite. The position of the G band is highly sensitive to the number of grapheme layers and it is visible at 1580 cm$^{-1}$. The G band position shifts to higher energy or higher wavenumber location as the layer thickness decreases [21]. The most intense of the D bands is the Di band, which is located in the first-order region at ~1350 cm$^{-1}$, and it is characteristic of unrecognized carbon. It is also known as a disorder or defect band and represents the mode of sp3 carbon atoms [22–24]. Another first-order band pertaining to structural disorder is the D2 band at ~1620 cm$^{-1}$ which can be observed as a shoulder on the G band. This shoulder becomes further developed in more disordered carbonaceous materials where the G band and D2 band merge, until a single feature is observed around 1600 cm$^{-1}$, which produces an apparent band broadening and upshifting of the G band [25].

The first-order Raman spectra were recorded from 900 to 1800 cm$^{-1}$, while the second-order spectra were recorded from 2400 to 3100 cm$^{-1}$. The peak positions, their height, full width of the peaks at half-maximum (FWHM), and area of the disorder peak (D) and order peak (G) in the first-order spectra were measured. Thus, the relative intensity ratio of D and G bands (R1 = D1/G) can be used as an indicator for the degree of graphite crystallinity [26–28]. Raman spectra of graphite can also be used as a geothermometer to estimate the peak metamorphic temperature in the rocks hosting graphite. Beyssac et al. [24] constructed a geothermometer, based on the two bands area ratio (R2), defined as (R2 = D1/(G + D1 + D2), which can be applied to regional metamorphism rocks, according to the formula T$_{Gr}$ (°C) = −445 R2 + 641. The geothermometer is valid for temperatures between 330 and 650 °C, and the uncertainty corresponds to ±50 °C. At higher temperatures, the R2 ratio remains fixed at ~0.05.

## 2.4. Beneficiation

### 2.4.1. Flotation

Graphite-bearing rocks from Rautalampi and Käpysuo were beneficiated by froth flotation to produce high-grade graphite concentrate. Flotation experiments were performed by GTK Mintec, Mineral Processing Pilot Plant, at Outokumpu. The samples were prepared by crushing and sieving to collect the sample fraction passing −1.4 mm. Crushed samples were then ground in one stage using a laboratory rod mill for 75 min (<250 μm). The d80 of the flotation feed samples was about 43 μm. The flotation feed C average head grade was 12.5%. In this study, the froth flotation experiments were conducted in a flotation machine, with cell volume 1.5 and 4 L. Rougher flotation was done to separate the graphite from their gangue minerals. Kerosene fuel ($C_{12}H_{23}$) was used as collector, and Flotanol 7026 plus methyl-isobutyl carbinol (MIBC, $C_6H_{14}O$) as flotation frothers for the graphite. Sodium silicate ($Na_2SiO_3$) and starch were used as depressants aiming to remove silicates and iron-bearing minerals, and the dosages were 1500 g/t and 450 g/t, respectively. The dosage of collector and frother varied within the tests, as presented in Table 1. Graphite separation was accomplished through several stages of cleaning since the objective of the study was to obtain high-grade graphite concentrates, which are suitable for refractory and battery applications. A schematic depiction of flotation circuits including initial stages of grinding and rougher flotation, followed by five cleaning stages, is given in Figure S1. The chemical analysis of the feed to the rougher concentrate and the final concentrate obtained is given in Table S3.

**Table 1.** Reagent types and dosage used for flotation test.

| Test Code | Main Variable | Reagent Dosages (g/t) | | | | |
|---|---|---|---|---|---|---|
| | | Flotanol 7026 | Kerosene | MIBC | Na$_2$SiO$_3$ | Starch |
| R7-1 | Test (PH19-9) | 204 | | | 1500 | |
| R7-2 | Reagents | | 220 | 345 | 1500 | |
| R7-3 | Lower collector dosage | | 111 | 172 | 1500 | |
| R7-4 | Starch | | 111 | 170 | 1500 | 450 |
| R7-5 | Rod Mill | | 116 | 187 | 1500 | |

The attritor mill (Union Process Model 1S) is mainly used for reducing particle sizes in the final graphite concentrate from the flotation processes. The milling was done using 3 to 4 mm diameter ceramic balls, 564 rpm, 3.8 L balls, and about 20% solids. After 360 min, the d50 was 24 μm. Additional grinding with a laboratory ball mill was made with various conditions. The first test run was done with a 5 kg mix of steel and ceramic balls, diameter 5–20 mm. Grain size degreasing appeared to proceed very slowly and milling was continued with steel balls, diameter 10–15 mm and weight 8 kg. Each suspension contained 350–400 g graphite concentrate and 0.5 L water.

### 2.4.2. Graphite Purification

Removal of silicate and gangue minerals from graphite ores was required to achieve high-purity graphite products. Alkaline roasting and acidic leaching process were used to prepare high-grade graphite (>99%) from fine flake graphite concentrates. These concentrates were products of multistage flotation-cleaning processes of Rautalampi and Käpysuo ore. The graphite purification method included several steps: alkaline roasting, water washing, sulfuric acid leaching, and drying to remove any impurities within the lattice structure help to attain a highly purified graphite [29–31]. The alkali roasting–acid leaching efficiency of pure graphite was investigated with the liquid–solid ratio of 2/1 (*w/w*) in alkaline roasting and 5/1 (*w/w*) in acidic leaching.

Firstly, the research samples of fine graphite were roasted with alkaline hydroxide (concentration NaOH 15% to 35%) at 250 °C; in this step, common impurities were converted to soluble forms. In the next step, the graphite was filtered, washed with water to remove residual alkalinity, and then dried at

105 °C. The roasted product was washed numerous times using deionized water until the washing solution reached natural pH = 7. After roasting, the graphite concentrate was treated with $H_2SO_4$ 10% concentration in a beaker for further removal of the insoluble compounds, mainly hydroxides and oxides. Finally, the resulting solution was filtered and washed several time with deionized water until pH reached neutral. Then, the mixture was dried and high-quality graphite powders (>99%) were obtained.

## 3. Results

### 3.1. Graphite Petrology and Mineralogy

Rautalampi and Käpysuo flake graphite is found mainly in two rock types: quartz–mica schist and feldaphitic biotite gneiss. The quartz–mica schist comprises the minerals quartz, feldspar (mainly plagioclase and orthoclase), and biotite as the main silicate minerals. The quartz and feldspar showed partly a granoblastic texture. Graphite and sulfide were the main opaque minerals. The graphite crystals most commonly occurred along the grain boundaries of other minerals and were often arranged parallel to other minerals, particularly biotite, and together they defined the foliation of the rock, forming a typical texture of the graphite schist (Figure 3a). Pyrite and pyrrhotite were the dominant sulfides and associated mainly with graphite and biotite (Figure 3b). The feldspathic biotite gneiss rocks consisted mainly of alternating quartz and feldspar bands (about 3 mm thick) and thin layers of biotite and graphite (about 0.5 mm) with subordinate rutile, garnet, and amphibole. Plagioclase feldspar was poorly to well twinned, occasionally myrmekitic and partially altered to sericite, whereas biotite was altered to chlorite (Figure 3a–d). The same replacement of plagioclase was also seen in the biotite schist.

The foliation-laminae in both rock types consisted predominantly of graphite flakes plus biotite and chlorite. The majority of the graphite flakes occurred as flat, plate-like crystals (>30 μm width), with angular and rounded edges, disseminated mainly in fractures and along the foliation. Graphite flakes in the studied samples ranged in size from 50 to 1600 μm in length. Commercial grade flake graphite can be subdivided into coarse flakes (400–1600 μm), medium flakes (150–400 μm), and fine flakes (<150 μm). From thin-section image analysis, most graphite flakes observed were oblong shaped, but not particularly fibrous, and the ratios between their long and short axes were in the range of 2 to 4 for the majority of the flakes (Figure 3c–f). The SEM images also showed that all the samples consisted of flaky graphite, that is, one graphite flake consisted of several layers, with regular and irregular flake edges and clean flake surfaces (Figure 3f).

All of the XRD patterns of ground graphite corresponded to highly crystalline hexagonal graphite. No peaks of rhombohedral graphite were recorded. The measured $d_{002}$ spacing showed no significant variation between the studied graphite samples (Figure S2). However, the measured full-width at half-maximum (FWHM) of the 002 peak was more sensitive to calculate the size of crystalline graphite. The *Scherrer formula* [32] is used to obtain the crystalline size along the c axis (*L*):

$$L = \frac{K\lambda}{\beta \cdot \cos\ \theta}$$

where β is the full width of the peak at half-maximum (FWHM) in radian, λ is the X-ray wave length in angstrom (Å), and θ is the angle of diffraction in radian. K is the shape constant, assumed to be 0.9 [33,34], the diffraction peak centered for studied graphite samples at 2θ value ranges (25.9–27.2 2θ°). The calculated crystallite size along the c axis (*L*) of studied graphite samples ranged between 130 and 150 nm with an average of 135 nm.

The XRF analysis and normative mineralogy are presented in Table 2, which shows the major constituents were $SiO_2$, $Al_2O_3$, $Fe_2O_3$, CaO, $K_2O$, MgO, $SO_3$, and $Na_2O$. In addition, the average carbon content was 12.5% FC (fixed-carbon content). The main gangue minerals were quartz, plagioclase, K-feldspar, biotite with subordinate pyrite, carbonate, and chlorite.

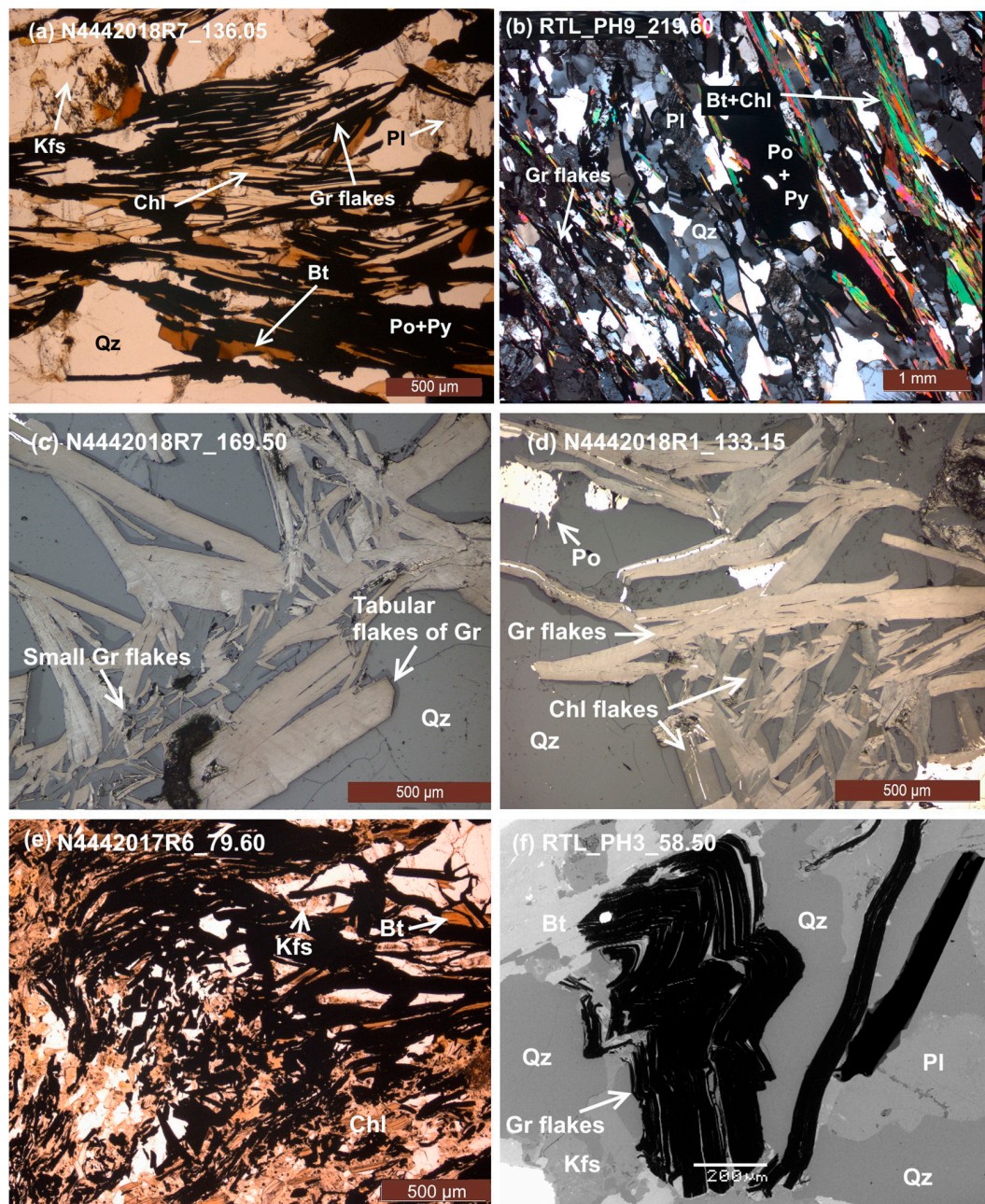

**Figure 3.** Petrography of a graphite-bearing schist and gneiss as seen in a polarization microscope. (**a**,**b**) The graphite flakes show strongly foliated schist consisting of alternating biotite, quartz, plagioclase, chlorite, and sulfide minerals; (**c**,**d**) reflected light image showing the large graphite flakes (≥1 mm) occurring as flat, plate-like crystals (>30 μm width), with angular and rounded edges; (**e**) plane-polarized light illustrating two graphite populations within one sample, coarse and fine graphite flakes; (**f**) selective graphite flakes as seen in SEM.

**Table 2.** Chemical composition and normative mineralogy of the studied graphite ore (wt. %).

| Composition | $SiO_2$ | $Al_2O_3$ | $Fe_2O_3$ | MgO | $TiO_2$ | CaO | $K_2O$ | $Na_2O$ | $P_2O_5$ |
|---|---|---|---|---|---|---|---|---|---|
| **Content** | 54.4 | 13.1 | 6.1 | 2.7 | 0.6 | 1.7 | 1.9 | 1.5 | 0.3 |
| **Composition** | BaO | SrO | $C_2O_5$ | $ZrO_2$ | CuO | ZnO | $Y_2O_3$ | $SO_3$ | C |
| **Content** | 0.05 | 0.02 | 0.03 | 0.02 | 0.01 | 0.007 | 0.002 | 7.3 | 12.5 |
| **Mineral** | Quartz | Feldspar | Mica | Carbonate | Pyrite | Chlorite | Apatite | Graphite | Others |
| **Content** | 35.4 | 25.4 | 15.5 | 4.3 | 4.5 | 1.1 | 0.6 | 12.5 | 0.5 |

## 3.2. Particle Size and Chemical Analysis

In the processing of graphite beneficiation, the crude ore (about 1–3 kg) was subjected to crushing first by using a jaw roll crusher to reduce it to the desired size (>1.4 mm), followed by stage grinding in rod or ball mills in closed circuit with classifier, before being sent to flotation. The particles produced, having different sizes and shapes, can be separated through a sieve series of screens: +1000, 700, 500, 250, 125, 180, 75, 45, and −20 μm. The weight percentage passing the sieve series was evaluated. The calculated d80 of the graphite samples N4442017_R2 and N4442017_R7 were found to be 850 and 930 μm, respectively. The carbon content of particles that passed through the sieve series was detected by Eltra analyzer (Table S4). The size distribution particles in the two representative samples N4442017_R2 and N4442017_R7 are shown in Figure 4a. The effect of particle size on carbon content in all size ranges was studied and is shown in Figure 4b. It is evident from the histograms that the highest grades of graphitic carbon were obtained at size ranges larger than the 250/125 μm fraction in sample N4442017_R2, while the highest carbon content in sample N4442017_R7 occurs in fraction +125–75 μm.

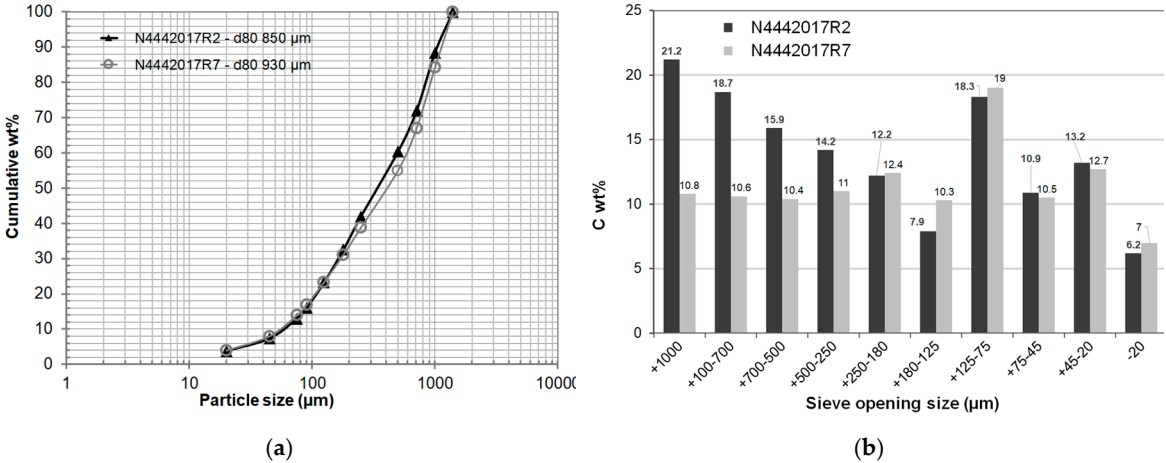

(a)　　　　　　　　　　　　　　　　　　　　　　　　　　　(b)

**Figure 4.** (**a**) Cumulative percentage against sieve sizes; (**b**) carbon content (%) of particle size distribution of graphite samples after crushing to particle sizes <1.4 mm (d80 = 850 and 930 μm for samples N4442017_R2 and N4442017_R7, respectively).

## 3.3. Raman Spectroscopic Characterization

Raman spectroscopy is a rapid and nondestructive technique that can be used to estimate the graphitization temperature and degree of crystallinity of the carbonaceous material (CM). Raman spectrum is sensitive to the imperfections in the graphite crystal structure, such as the lattice defects, finite size of crystallites, and edges of graphite layers. Beyssac et al. [24] derived the first temperature-dependent empirical equation, in order to estimate the peak temperature between 330 and 650 °C during metamorphism. This was a developmental turning point and encouraged the wider use of Raman spectroscopy as a geothermometer for graphite-bearing rocks. In order to quantify the observations for Käpysuo and Rautalampi graphite samples, spectral parameters were determined by background fitting process and the corresponding dataset is given in Figure 5. The most widely used parameters obtained from Raman spectra were obtained from 35 petrographic thin sections of graphite-bearing rocks, which are summarized in Table 3. To achieve wider utilization of Raman, the spectrum processing method, and the positions and nomenclature of Raman bands and all parameters including mean values for center position, FWHM of the D1 and G bands, values of the D1/G intensity and peak area (D1/(G + D1 + D2)) ratios are required to be standardized. An assessment of the most widely used Raman parameters, as well as the best analytical practices that have been proposed, was conducted. Based on these quantitative parameters, it is possible to evaluate the metamorphic condition of graphite-bearing samples, according to the formula $T_{Gr}$ (°C) = −445 R2 + 641. The application of $T_{Gr}$ to the studied samples gave a temperature range of 430–560 °C.

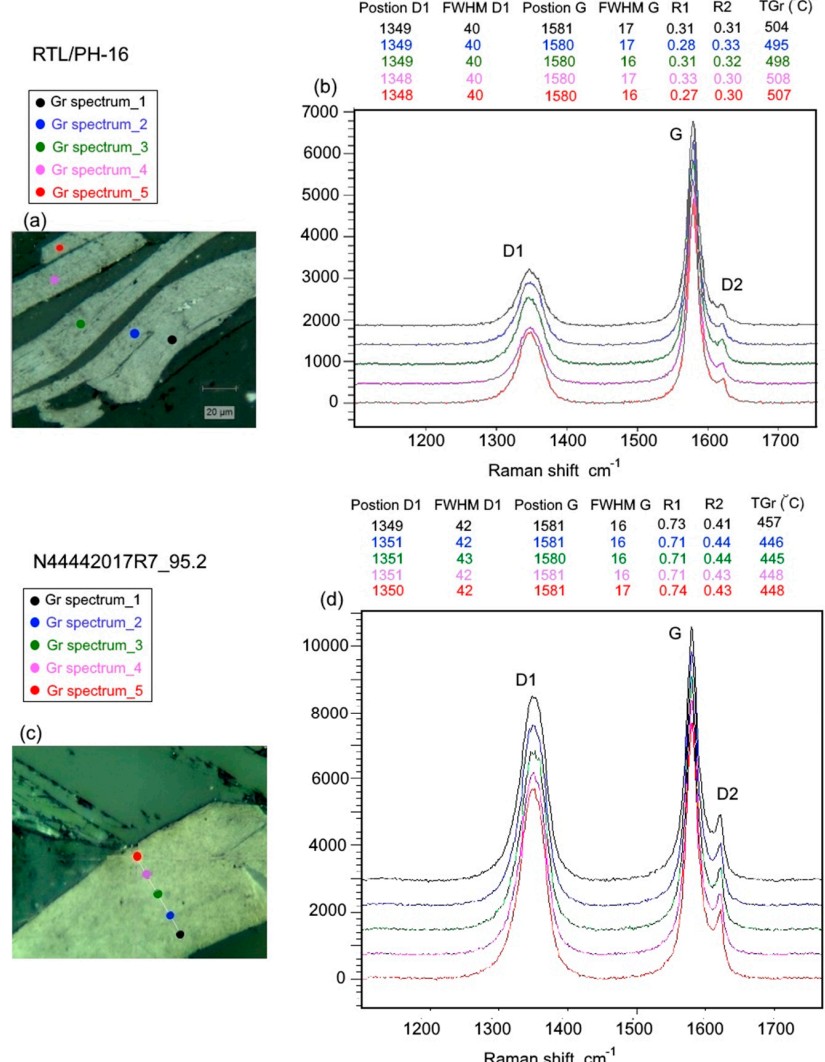

**Figure 5.** (**a**,**c**)Optical microscope images of graphite flakes from two samples, RTL_PH16 and N4442018_95.7; (**b**,**d**) Raman spectrum of single graphite flakes. The color of Raman spectrum corresponds to the color of the spot where the spectrum was acquired.

**Table 3.** Raman spectra parameters from petrographic thin section and the peak of the metamorphic temperature, according to the formula of Beyssac et al., * $T_{Gr}$ (°C) = −445 R2 + 641 [24].

| Samples | Peak Position | | FWHM | | R1 | R2 | $T_{Gr}$ (°C) * |
|---|---|---|---|---|---|---|---|
| | $D_1$ | G | $D_1$ | G | | | |
| High-crystalline graphite | | | | | | | |
| RTL_PH3_52.95_2 | 1352 | 1580 | 37 | 14 | 0.10 | 0.20 | 560 |
| M2143_98_R330_67.3_2 | 1350 | 1580 | 40 | 16 | 0.29 | 0.29 | 514 |
| RTL_PH11_58.85_4 | 1350 | 1580 | 41 | 16 | 0.30 | 0.30 | 505 |
| N4442018R30_54.05_2 | 1349 | 1581 | 40 | 16 | 0.45 | 0.32 | 498 |
| M4121_61_R1_11.80_1 | 1350 | 1580 | 40 | 17 | 0.45 | 0.38 | 470 |
| Low-crystalline graphite | | | | | | | |
| RTL_PH3_152.0_1 | 1350 | 1580 | 42 | 17 | 0.85 | 0.48 | 420 |
| RTL_PH3_152.0_3 | 1349 | 1580 | 42 | 17 | 0.90 | 0.48 | 430 |
| RTL_PH9_238.0_2 | 1350 | 1580 | 40 | 17 | 0.68 | 0.46 | 434 |
| RTL_PH8_155.85_1 | 1351 | 1580 | 41 | 16 | 0.65 | 0.46 | 436 |
| RTL_PH6_127.6_1 | 1351 | 1580 | 41 | 17 | 0.70 | 0.44 | 440 |

Wide ranges of intensity ratio (R1) and peak area ratio (R2) were recorded for the graphite flakes in 35 thin sections and polished sections from selected samples ranging from 0.09 to 0.9 for R1 and 0.18 to 0.48 for R2 ratio (Table 3), indicating that the graphite flakes formed under high- and low-temperature metamorphism, respectively. As the degree of graphitization increased, the Raman spectra of the graphite G band became narrower, and the D1 band appeared broad as graphitization increased, where the Raman spectra of the graphite band (G band) became narrower, and the D1 band appeared as a broad band with lower relative intensity than those of low-grade graphitization [35,36].

In the high-crystalline graphite flakes samples, the R1 and R2 ratios decreased down to 0.1–0.45 and 0.2–0.38, respectively. The Raman spectra flakes showed a strong G band with quite broad D1 band and almost undetectable D2 band (Figure 5a,b). It is referenced that fully ordered graphite in the studied rocks did not appear until metamorphism conditions were reached at temperatures exceeding 450 °C as recorded in the range of 470–560 °C.

Another interesting observation was the comparison between the evolution of R1 and R2 in the low-crystalline graphite flakes (disordered graphite); the R1 intensity ratios (0.65–0.90) and R2 peak area ratios (0.44–0.48) showed higher values than those obtained for the somewhat high-grade crystalline graphite flakes in the former samples. The disorder-induced D1 and D2 bands are observed in Figure 5c,d which shows progressively better defined D1 and D2 bands than those of the high-grade graphite samples and exhibits higher values of both intensity ratios R1 (D1/G) and peak area ratios R2 [D1/(G + D1 + D2)]. We refer to this type of disordered graphite formed at lower temperature recorded range from 400 to 440 °C. Recently, Palosaari et al. [37] determined the peak metamorphism of graphite occurrences in Piippumäki, Eastern Finland using Raman spectrum was 737 °C.

The presence of both ordered and disordered graphite in the studied samples can thus be explained by a superimposition of more than one metamorphic event. In addition, disordered flakes intersecting most metamorphic mineral assemblages could clearly be distinguished as having formed later than ordered graphite flakes. The presence of ordered and disordered graphite can thus be explained by a superimposition of more than one metamorphic event. We report here thermometer plots of the calculated dependence on both R1 and R2 representing the used graphite flakes, and calibrated using the formula of Beyssac et al. (RSCM) [24]. These datasets have demonstrated a linear correlation in R2 values over the range 420 to 520 °C, but there was little variation in R1 values at low temperature fields (Figure 6a). The graphite crystallinity was dominated and strongly correlated with peak metamorphic temperature. The results of Figure 6b show that the center positions of D1 bands and G bands shifted slightly with increasing metamorphic temperature from 1352 to 1348 cm$^{-1}$ and 1581 to 1579 cm$^{-1}$, respectively.

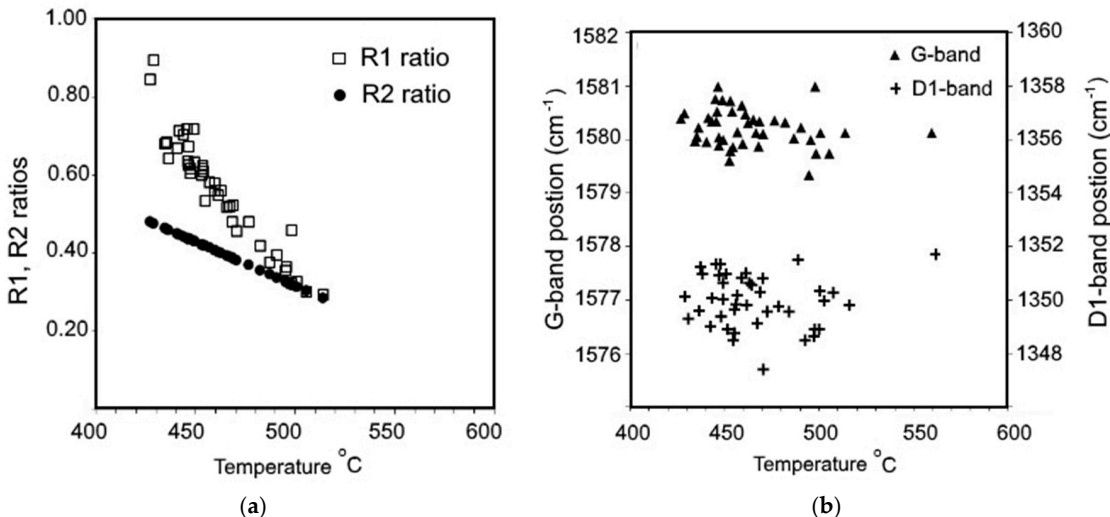

(**a**)　　　　　　　　　　　　　　　　　　　　　　　(**b**)

**Figure 6.** Temperature measurement results from Raman spectroscopy such as: (**a**) intensity ratio (R1) and area ratio (R2); (**b**) D1 and G bands positions.

*3.4. Graphite Beneficiation*

3.4.1. Flotation

Graphite ore is mostly concentrated from crushed rocks by using flotation separation techniques. The beneficiation processes depend upon the nature and association of gangue minerals present in ore deposits [38–40]. The mineralogical results indicated that the flaky graphite particles from Rautalampi and Käpysuo deposit were embedded with gangue minerals such as quartz, mica, sericite, clay, and sulfide minerals, making it difficult to be beneficiated by using the typical flotation technique. Gangue mineral of sericite, mica, and clay have a similar flake or scaly form to graphite, and clay shows good adhesion on the surface of graphite, so flotation of the graphite is antagonistically affected significantly. Dissociation between graphite and sulfides has certain difficulties; especially fine graphite particles filled in the pore of pyrite and pyrrhotite are difficult to dissociate, and this may have a certain influence on the graphite concentrate quality. A combination technique of two rougher flotations at natural pH and five stages of cleaner flotation was used to increase the graphite recovery based on the coexisting relationship between graphite and gangue minerals in middling. The ore contained 5 to 20 wt. % C with an average of 12.5 wt. % C (Tables S1 and S2). By two rougher flotation stages at natural pH, the grade graphite was increased to 51–57% fixed carbon. This product was the feed to the flotation cell; five stages of cleaner flotation are essential to produce clean graphite concentrates of 80–86% total carbon content with 67–97% total carbon recovery.

The results of graphite grade (FC) and recovery achieved for the rougher flotation and final graphite concentrate (CC5) sample R7 are summarized in Table 4 and Figure 7. All of the cumulative grade-recovery curves were very similar and exhibited similar gradients, except one for sample R7-4 showed a clear difference with a steeper slope of the grade-recovery curve. Based on flotation tests of composite sample R7, the best grade of final concentrate containing 85% fixed carbon at 88% recovery was achieved in the test R7-5, where the treatment was done with a pH of 8.5 and the chemical dosages were 187 g/t MIBC, 116 g/t kerosene, 1500 g/t sodium silicate, as presented in Table 1. The graphite grade of the final concentrate upgraded from 57% to 85%, indicating that the two rougher flotations followed by five stages of cleaning treatments can efficiently recover the graphite concentrate for the battery industry.

**Table 4.** Results of flotation tests after fifth cleaning stage (CC5).

| Flotation Stages | Rougher Flotation | | Final Phase Cleaning CC5 | | | Combined Tailings |
|:---:|:---:|:---:|:---:|:---:|:---:|:---:|
| Test Code | C Grade % | C Recovery % | C Grade % | C Recovery % | Mass Pull % | C Grade % |
| R7-1 | 57 | 98.4 | 86 | 70 | 8.5 | 3.4 |
| R7-2 | 50.54 | 99.2 | 80 | 98 | 12.6 | 0.3 |
| R7-3 | 53.28 | 98.9 | 84 | 79 | 9.6 | 2.4 |
| R7-4 | 57.61 | 97.2 | 84 | 67 | 10.3 | 3.8 |
| R7-5 | 56.74 | 97.6 | 85 | 88 | 10.7 | 1.4 |

3.4.2. Graphite Flake Size Reduction

The common techniques for the preparation of micron-size graphite are ball, attritor, or jet mill. In this study, additional milling was carried out by using both the attritor and ball mill. The maximum grade of the final (5th cleaner) concentrate was obtained at +125–250 μm fraction, having 97% FC. The FC content of the final concentrate in the finer fraction decreased continuously as 90.7% C for the 75–45 μm fraction and 82.6% C for the –20 μm fraction. The particle size of the final concentrate d50 = 60 μm was reduced to 20 μm. The SEM images show that the final graphite concentrate consisted of fine crystalline flaky graphite with 10~15 μm thickness and 50–100 μm of width. Each graphite flake was composed of several layers with irregular and irregular flake edges and clean flake surface (Figure 8a,b). The treatment is not expected to have modified the graphite flakes' microstructure.

As expected, the general flat shape of the graphite flakes is still visible (Figure 8) and the high crystallinity was indicated by X-ray diffraction (XRD) patterns (Figure 2 and Figure S2).

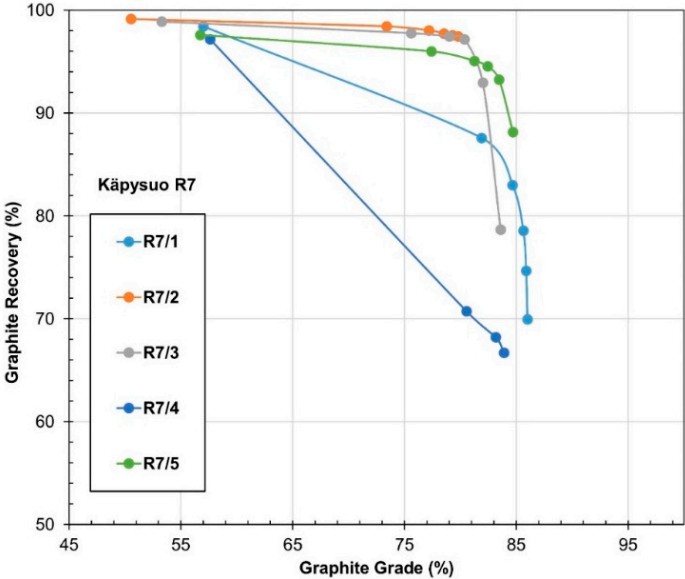

**Figure 7.** FC grade and recovery curves graph for the five R7 flotation tests. The points represent the FC grade and recovery in each flotation stage. The first points in the curves represent the rougher flotation and the following ones, the cleaners.

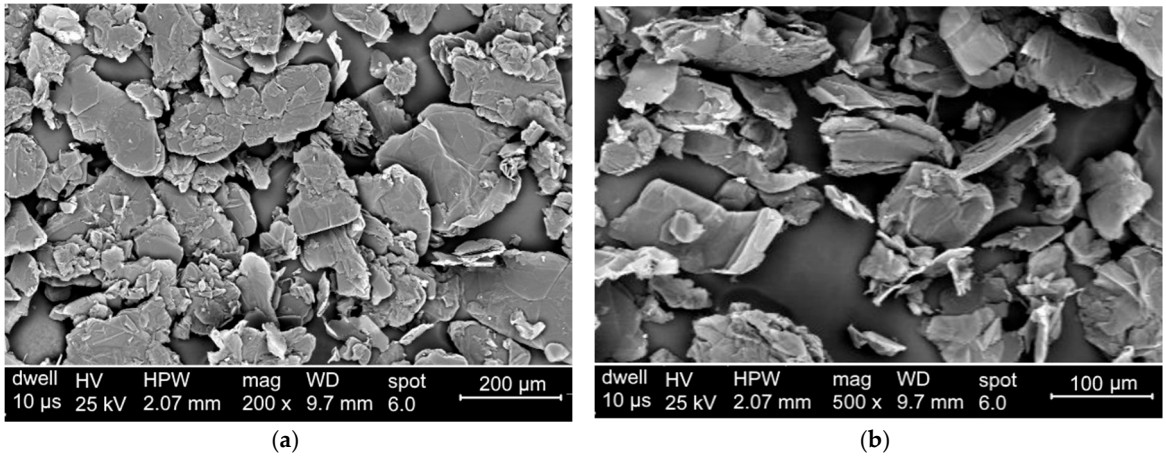

**Figure 8.** Scanning electron microscope (SEM) images showing the microstructure of graphite flakes after beneficiation processes at: (**a**) low magnification; (**b**) highest magnification.

### 3.4.3. Graphite Purification

Alkali treatment with graphite concentrate was mostly related to the concentration of alkali. It can be observed that increasing the NaOH concentration from 15% to 35% improved substantially the C grade in the residue as well as that the further raise of the concentration to higher than 35% did not improve any more the process (Figure 9a).

The effect of alkaline roasting temperature on the removal of impurities from graphite concentrates is quite clear. The increase of temperature leads to increasing the removal of impurities, especially at the temperature range of 150–200 °C. The product purity reaches 93.5% at 150 °C and increases peak value of 99.4% at 200 °C from a feed graphite powder purity of 97.7% (Figure 9b). In addition to this, the effect of leaching time on the removal of impurities from graphite was studied. It seems that increasing in leaching time has a positive effect on removing impurities. The removal of impurities

increased rapidly by increasing leaching time from 30 to 120 min, but the leaching time seemed to have an insignificant effect over 120 min (Figure 9c).

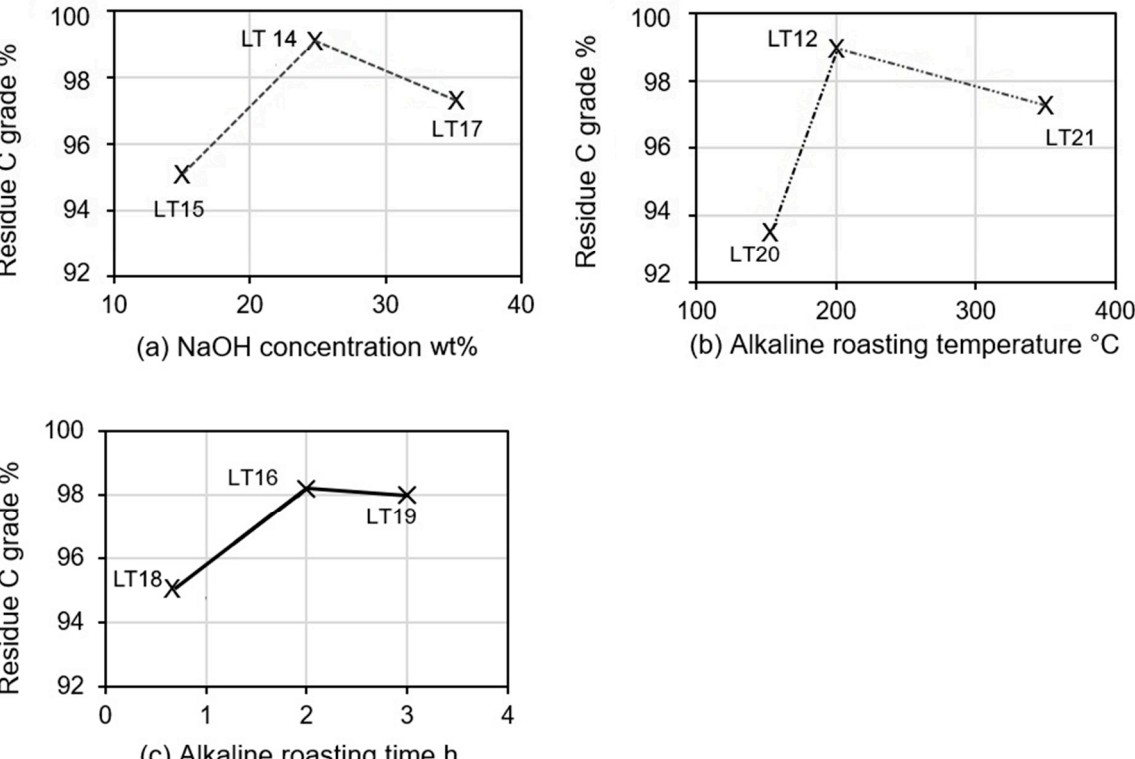

**Figure 9.** Effect of roasting and acid leaching process on graphite purity; (**a**) high-grade graphite at 200 °C and for 3 h leaching time; (**b**) alkaline roasting temperature at 3 h concentration time; and (**c**) roasting time at 250 °C.

Both alkaline roasting temperature and alkaline roasting time are equally good parameters for eliminating impurities in fine graphite powder. Increasing the time and temperature will often increase the rate of reaction of silicate and sulfide impurities. For instance, the concentration of $SiO_2$ in the final product could be reduced to less than 0.4% from 4.0% and sulfur content reduced to below 0.05% from 0.23% at 220 °C. The decrease of sulfur from 0.50% to 0.07% was most effective at a temperature of 350 °C.

Alkaline purification of the studied graphite showed excellent results as well. The results prove it is possible to reduce impurities 1.5–1.7 wt% and the graphite content can be increased from 93.5% to reach up to 99.4% carbon content, as shown in the samples tested, LT14 and LT 12, respectively (Table S4). That purity was considered adequate for the first experiments as anode active graphite in lithium-ion batteries.

## 4. Conclusions

1. Rautalampi and Käpysuo graphite ores mainly consist of quartz–mica schist and feldspathic biotite gneiss. These rocks are associated with metamorphic indicator minerals as chlorite, garnet, and sillimanite. The graphite flakes ranged in size from 50 μm to 1600 μm, but with the majority ranging from 200 to 500 μm. Coarse graphite flakes occur as flat, plate-like crystals (≥30 μm width), with angular and rounded edges, associated mainly with chlorite, biotite, and iron sulfide minerals. Small graphite flakes were occurring as fracture, cavity-filling, or fissure-filling in veins. However, some graphite flakes were concentrated along the mineral boundaries. Impurities in graphite ore and

its surrounding host rocks were quartz, mica (biotite and chlorite), and sulfide minerals (pyrite and pyrrhotite).

2.   Raman spectroscopy study of graphite flakes provide, relatively easily and quickly, comprehensive information on the degree of graphitization as a function of microstructure and heat treatment, which may contribute to determine the maximum temperature reached during regional and contact metamorphism. The formation temperature of ordered graphite flakes in Rautalampi and Käpysuo was in the range of 470–590 °C, whereas that for the disordered graphite flakes ranged from 400 to 440 °C, which was related to the condition of retrograde metamorphism.

3. The crushed <1.4 mm graphite ore from the selected drill core sample was further ground with rod mill grinding and then screened down to 250 μm (d80). The ground ore was concentrated by two stages rougher flotation, followed by five cleaning stages. The final concentrate presented 85% fixed carbon at 88% recovery. The flotation was done with a pH of 8.5 and the chemical dosages were 187 g/t MIBC, 116 g/t kerosene, 1500 g/t sodium silicate.

4.   Further purification by alkaline roasting process was still needed to produce high-grade graphite with up to 99% carbon content to be used in the battery industry. The leaching process for producing high-purity graphite from 87–93.5% to 99.4% carbon content was studied by using many factors, such as concentrated sodium hydroxide (NaOH, 15–35%) at 250 °C, sulfuric acid concentration $H_2SO_4$ of 10%, water washing, and finally drying.

5. The results showed that the graphite content of Käpysuo ore can be increased with alternative purifying methods like leaching chemicals (hydrochloric acid, hydrofluoric acid) into ultra-high-purity graphite (≥99.95% C) with fine particle sizes <20 μm, which reaches the requirement for use in lithium-ion battery testing.

**Supplementary Materials:** The following are available online at http://www.mdpi.com/2075-163X/10/8/680/s1, Table S1: Samples selected for mineralogical and beneficiation processes, Table S2: Whole rock geochemistry, major oxides (wt. %), trace elements (ppm), and carbon content based on Eltra measurement, Table S3: Chemical analysis of feed material and flotation test products, C content by Eltra (%) and element contents by XRF (%), Table S4: Result of sieving analysis and corresponding FC content in wt. %, Figure S1: Schematic flow sheet of graphite flotation. R7-5 is used as an example. The conditioning time for kerosene and MIBC was 2 min in all stages, Figure S2: X-ray diffraction patterns of high-grade graphite products for several samples.

**Author Contributions:** T.A.-A. and S.L. designed the concept of the manuscript, T.A.-A. performed mineralogical and Raman studies, D.S. carried out graphite purification processes, S.L. and T.A. contributed to the sample selection, preparation, and image processing, and all the authors contributed equally in the discussion of the results and writing the article. All authors have read and agreed to the published version of the manuscript.

**Funding:** This research was funded by BUSINESS FINLAND, grant number 7636/31/2017.

**Acknowledgments:** The authors would like to acknowledge all the members in GTK mineral research laboratory and particularly Bo Johanson, Lassi Pakkanen, and Mia Tiljander. We also thank GTK Mintec, Mineral Processing Pilot Plant for assistance in graphite beneficiation processing. Thanks to BUSINESS FINLAND for their funding and support.

**Conflicts of Interest:** The authors declare no conflict of interest.

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
