# Peer review of "High-Grade Flake Graphite Deposits in Metamorphic Schist Belt, Central Finland—Mineralogy and Beneficiation of Graphite for Lithium-Ion Battery Applications"

_minerals, doi:10.3390/min10080680_

Round 1

Reviewer 1 Report

First, the title and abstract of this review request seem to be different from the manuscript provided.

The paper is interesting and especially the processing mineralogy part is well written and provides important new data. The description of the geological background and samples themselves is rather limited.

My main concerns are:

  • The english of some chapters requires improvement.  I have only partly corrected the english into the annotated file attached here.
  • Discussion of metamorphic history need to be extended:
    • how the obtained temperatures compare with those obtained from surrounding garnet-cordierite gneisses. Even the highest temperatures recorded by graphite thermometer are far from granulite facies, so could these rocks be called as high grade metamorphic gneisses?
    • The authors suggest multiple metamorphic events to explain the range in temperatures. Also the possiblity of retrogade metamorphism need to be discussed. It has recently been shown that highly crystalline graphite can metastably preserve to lower P-T conditions (Palosaari et al., Mineralium Deposita 2020, published online).

Additional comments in the annotated m/s as attached

Author Response

Response to Reviewer 1 Comments

Please see below, in red, our detailed response to comments.

Reviewer 1 Comments

Point 1 First, the title and abstract of this review request seem to be different from the manuscript provided.

Response 1: Thanks for your comment, the manuscript is dealing with the mineralogical studies of graphite-bearing rocks (petrography, XRD and SEM) and beneficiation processing (multistage flotation processes, alkaline-roasting and acid leaching) of these rocks to obtain high purity graphite for lithium batteries applications.

Point 2 The paper is interesting and especially the processing mineralogy part is well written and provides important new data. The description of the geological background and samples themselves is rather limited.

Response 2: Thanks for your interest in our work, we updated the introduction

My main concerns are:

- The english of some chapters requires improvement. I have only partly corrected the english into the annotated file attached here. English language of the manuscript was improved.

- Discussion of metamorphic history need to be extended: how the obtained temperatures compare with those obtained from surrounding garnet-cordierite gneisses. Even the highest temperatures recorded by graphite thermometer are far from granulite facies, so could these rocks be called as high grade metamorphic gneisses?

Response 3: In this manuscript we dissuaded explain only the Raman spectrum of graphite flakes can therefore be used as a geothermometer of the maximum temperature conditions of the graphite-bearing rocks. We are currently preparing of such case studies´´ Graphite thermometer determined by a Raman spectroscopy compare with P-T conditions estimated by garnet-biotite thermometer``. Also according the individual bulk rock compositions of the graphite-bearing rocks, we will calculate P–T pseudo sections using THERMOCALC 3.33 and obtain phase equilibria modelling for discussing P-T history of the surrounding rocks in Rautalampi and Käpysuo.

- The authors suggest multiple metamorphic events to explain the range in temperatures. Also the possiblity of retrogade metamorphism need to be discussed. It has recently been shown that highly crystalline graphite can metastably preserve to lower P-T conditions (Palosaari et al., Mineralium Deposita 2020, published online).

Response 4: According to Palosaari et al., 2020 for flake graphite occurrence in Piippumäki, Eastern Finland, the peak metamorphic temperature of 737 °C (is higher not lower as reviewer mentioned) was determined by a Raman thermometer, and no temperatures of greenschist facies were observed. A pseudosection was constructed from whole-rock chemical composition and indicated equilibration at ca 5 kbar and 740 °C, which corresponds to the observed mineral assemblages.

Reviewer 2 Report

The manuscript has many issues pertaining to its structure, editing and in general the English language. The paper requires major efforts from authors to restructure the sections, eliminate unnecessary and repeated information. The paper can be shorten significantly by careful and thoughtful organization and edits to make section coherent and easy to follow. For example -

a) The figures and tables should be close to the text where authors are describing them. Some tables and figures are essentially providing same information and add unnecessary length to the paper.

b) The materials (equipment) and methods are repeated/or provided in the results section rather all should be in section 2. 

c) In section 3.4.1, paragraphs are repeated multiple times (particularly related to Attrition Mill work description) with exact wording. 

d) There are many grammatical errors (tenses) and spellings errors at multiple places. 

Apart to apparent aforementioned issues, there are some technical mistakes that the authors must address before the manuscript should be considered for publication. 

e) Introduction - 

i) Must provide some background on typical lithium-battery industry, graphite consumption, need, acceptable purity level etc. 

ii) How typically graphite is commercially graded (only on the basis of size?)

iii) Page 2, paragraph "in this paper..."should move at the end of the introduction reflecting the motivation for this research work.

iv) Need to add some background on flotation chemistry of this particular ore. Purpose of depressant (for sulfide minerals?)  Reason to test starch as typically it will depress graphite (which results shows as well for test R7/4)

f) Methods and Material - 

i) Fig. 2(a) and 2(b) are not clear. Suggest showing some details such as schist, gneiss bands, and graphite for easy to understand.

ii) Figure 4 - It will be very helpful if authors can show reagent dosages, conditioning and retention time at every step of rougher-cleaning stages. Further, what happened to the rougher/cleaner tails. Were they combined together? What was the final tail graphite content? How authors determine optimal reagent dosage? 

iii) Graphite in general is a naturally floatable material, why a test was not considered without collector. It would be interesting if collector addition really make any difference in the recovery.  

iv) Have any studies conducted by grinding rougher concentrate prior to treat in the cleaner stages?

g) Results - 

i) Section 3.2 - Repetitive information on screening size and all. The top size should be 1000 microns. D80 passing size indicated is conflicting with that shown in Figure 7(b). Table 4 is not needed. Figure 7(a) & (c) and 7(b) & (d) can be combined. 

ii) It is not clear at what stage Attrition mill was used? Is it prior to the leaching process?

iii) What are sample ID LT#No. indicates? Are they sub-samples of same flotation product?

iv) Figure 10, circles for rougher grade/recovery data are not clear. Discussion is needed why we see variability in test results for R7/1 to R7/5. What was the feed carbon content for these tests (after rod mill grinding?) May be adding columns in Table 9 showing head grade and total tail grade will be helpful.

Author Response

Response to Reviewer 2 Comments

Please see below, in red, our detailed response to comments.

Reviewer 2 Comments

The manuscript has many issues pertaining to its structure, editing and in general the English language. The paper requires major efforts from authors to restructure the sections, eliminate unnecessary and repeated information. The paper can be shorten significantly by careful and thoughtful organization and edits to make section coherent and easy to follow. For example

a) The figures and tables should be close to the text where authors are describing them. Some tables and figures are essentially providing same information and add unnecessary length to the paper. According to your comment, we removed some unnecessary tables and figures

b) The materials (equipment) and methods are repeated/or provided in the results section rather all should be in section 2. We removed duplicate text

c) In section 3.4.1, paragraphs are repeated multiple times (particularly related to Attrition Mill work description) with exact wording. We removed duplicate text and we also updated the Attrition Mill work description.

d) There are many grammatical errors (tenses) and spellings errors at multiple places. The English grammar was approved.

Apart to apparent aforementioned issues, there are some technical mistakes that the authors must address before the manuscript should be considered for publication.

e) Introduction -

i) Must provide some background on typical lithium-battery industry, graphite consumption, need, acceptable purity level etc. According to your comment we updated the introduction.

ii) How typically graphite is commercially graded (only on the basis of size?); Done

 iii) Page 2, paragraph "in this paper..."should move at the end of the introduction reflecting the motivation for this research work. Done

iv) Need to add some background on flotation chemistry of this particular ore. Purpose of depressant (for sulfide minerals?) Reason to test starch as typically it will depress graphite (which results shows as well for test R7/4). We updated the 2.4.1. Flotation section according to reviewer comment. 

f) Methods and Material -

i) Fig. 2(a) and 2(b) are not clear. We removed Fig. 2 from manuscript.

ii) Figure 4 - It will be very helpful if authors can show reagent dosages, conditioning and retention time at every step of rougher-cleaning stages. Further, what happened to the rougher/cleaner tails. Were they combined together? The cleaner tails were kept separately.

What was the final tail graphite content? For example, for R7-5, the tail graphite content was 1.4%.

How authors determine optimal reagent dosage? The optimal dosage was determined based on a suitable C grade and recovery. In other words, a dosage that produce a high-grade carbon concentrate and with high C recoveries.

iii) Graphite in general is a naturally floatable material, why a test was not considered without collector. It would be interesting if collector addition really make any difference in the recovery. The R7-1 test was done just using the Flotanol C7 frother, so without any collector.

iv) Have any studies conducted by grinding rougher concentrate prior to treat in the cleaner stages. Yes. Although it was not seen significant improvement in the C grade. In addition, that lead to more C losses during cleaner stages. Yes see the references [23-25].

g) Results -

i) Section 3.2 - Repetitive information on screening size and all. The top size should be 1000 microns. D80 passing size indicated is conflicting with that shown in Figure 7(b). Table 4 is not needed. Figure 7(a) & (c) and 7(b) & (d) can be combined. Done

ii) It is not clear at what stage Attrition mill was used? Is it prior to the leaching process? It is used to reduce solid particle size.

iii) What are sample ID LT#No. indicates? Are they sub-samples of same flotation product? They are floatation products, but after alkaline roasting and acid leaching treatments.

iv) Figure 10, circles for rougher grade/recovery data are not clear. Discussion is needed why we see variability in test results for R7/1 to R7/5. What was the feed carbon content for these tests (after rod mill grinding?) May be adding columns in Table 9 showing head grade and total tail grade will be helpful. According to Reviewer comment, the Figure 10 was updated to 6 and Table9 was updated to Table 6

Reviewer 3 Report

This paper is of interest to the industry and researchers alike.  However, it is very long and somewhat tedious to read.  The authors MUST rework the manuscript such that it is not more than 10 or 11 pages long, including figures and tables.  They MUST be more succinct, clear and concise.

The material and methods section can be significantly reduced.  For example, in the chemical analysis section all you need say is that the "The samples were analysed at Eurofins Labtium Oy, Finland using XRF and ICP-MS."  Results of these analysis should be in the results section.

The tables contain a huge amount of data.  Is it necessary to present it all?  For example, Table 1 obviously has significant to the authors, but is it relevant to the reader?  I don't need to know about the drill core number, depth of thin section number.  Think carefully about the overall story you are trying to tell and your audience . . . what are the points you want then to take away.

Table 2 also contains lots of information, and if i compare the four columns the assays are similar for all four samples.  Perhaps a general comment of the typical composition is all that is needed.

The first paragraph after the title 3.Results should be deleted as it is an artifact of the template used to write the paper.

What does "deformed together" mean?  I think the mineralogical/metallurgical term would be locked as opposed to liberated.

Some of the figure cations are very lengthy.  Please shorten.

Table 4 is a little confusing.  I assumed that the two weight percent columns correspond to the two carbon content columns in the same order?

The figures and tables tend to be scattered through the paper and quite often appear several pages after being discussed in the text.  This tends to make it more difficult and confusing to read.

I could go on, but I think you get the picture . . . simplify and be concise.  Think of our audience.

Author Response

Response to Reviewer 3 Comments

Please see below, in red, our detailed response to comments.

Reviewer 3 Comments

Point 1This paper is of interest to the industry and researchers alike. However, it is very long and somewhat tedious to read. The authors MUST rework the manuscript such that it is not more than 10 or 11 pages long, including figures and tables. They MUST be more succinct, clear and concise. According to your comment, we removed some unnecessary tables and figures.

Point 2: The material and methods section can be significantly reduced. For example, in the chemical analysis section all you need say is that the "The samples were analysed at Eurofins Labtium Oy, Finland using XRF and ICP-MS." Results of these analysis should be in the results section. Done

Point 3: The tables contain a huge amount of data. Is it necessary to present it all? For example, Table 1 obviously has significant to the authors, but is it relevant to the reader? I don't need to know about the drill core number, depth of thin section number. Think carefully about the overall story you are trying to tell and your audience . . . what are the points you want then to take away. We crossed Table 1 from manuscript.

Point 4: Table 2 also contains lots of information, and if i compare the four columns the assays are similar for all four samples. Perhaps a general comment of the typical composition is all that is needed. Table 2 contain only four samples been analysed and each sample has differ carbon and elements contents.

Point 5 The first paragraph after the title 3.Results should be deleted as it is an artifact of the template used to write the paper. Done

Point 6 What does "deformed together" mean? I think the mineralogical/metallurgical term would be locked as opposed to liberated. Done

Point 7 Some of the figure cations are very lengthy. Please shorten. Done

Point 8 Table 4 is a little confusing. I assumed that the two weight percent columns correspond to the two carbon content columns in the same order? I think they are not the same, please reread the table again.

Point 9 The figures and tables tend to be scattered through the paper and quite often appear several pages after being discussed in the text. This tends to make it more difficult and confusing to read. We rearranged the tables and figures.

I could go on, but I think you get the picture . . . simplify and be concise. Think of our audience. Thanks for your comments.

Reviewer 4 Report

The paper reports on a set of laboratory experiments for the study of “High-grade flake graphite deposits in metamorphic schist belt, Central Finland-Mineralogical and beneficiation of graphite in the lithium ion batteries application”

It is an interesting work, however it can only be accepted with minor changes.

I have listed some comments below

Page 1, Line 40

and ≥75 µm      must be?     and ≥45 µm

Page 2 line 55

and –volcanic sequences     must be     and volcanic sequences

page 4, Table 1

N4442017R2 and N4442017R7;  You did not show the weight of these two samples. What is the importance of showing the weight?

Was  the N4442017_R7 sample the only one subjected to flotation tests?

Fig 4

Do CT1, CT2, CT3, CT4 and CT5 products return to the previous cell?

Are the reagents, including frother, only added at the beginning of the process, before the two stages of rougher?

Line 245

screens; +10000 µm     must be     screens; +1000 µm

Does Table 4 show the results for the size fractions?: +1400-1000? µm; +1000-710?; +710-500;    +45-20 and <20 µm.

Does Figure 7 also show the results for the size fractions: +1400-1000? µm; +1000-710?; +710-500;    +45-20 and <20 µm.

Does Figure 7 show the cumulative of the content of C?

Are figures 7a and 7b correct? They represent the results (cumulative) of the content? (shown in Table 4) In Figure 7b, D80 is 100 µm and not 930 µm

Table 4 and Figure 7 show different results for the carbon content in the fractions +45-20 µm and <20 µm. For example, for the sample N4442017_R7, in table 4 those fractions present carbon contents of 10.5% and 19.56% and in Figure 7d the contents are 23.2% and 6.9%.

Page 10

Line 251

“It is evident from the histograms that the highest grades of graphitic carbon were obtained at size ranges larger than the 250 µm fraction in both samples. High carbon content occurs also in fine fraction +45-20 µm in sample N4442017_R7.”

Is this correct? Or…..

It is evident from the histograms that the highest grades of graphitic carbon were obtained at size fraction of +250/125 µm.

But, based on table 4: High carbon content occurs also in fine fraction (<20 µm) for two samples

Has the effect of particle size on flotation been analyzed?

What is the effect of entrainment on the quality of the floated?

Page 13  line 327

 “This product was the feed to flotation column, five stages of cleaner flotation…”

In the cleaner flotation did you use a column or a cell?

Change order of tables 8 and 9 

(Table 9 is referred in the text before table 8)

(Line 330:     Table  9;        Line 386   Table 8)

Table 3;  and line 336

What is Flotanol? Not described in the sub-chapter: material and methods

The use of only three wash flotation stages could lead to better results. Although the use of many (five) flotation stages increases the content of the float, it strongly penalizes recovery.

Although stages 4 and 5 allow to increase the content, they caused a marked decrease in recovery. For example, for sample R7-3, for 5 stages the content is 84% and the recovery is 78.5%, but if you only used 3 stages you would get a content of 80.5% and a recovery of 97.5%

Line 339

Text between line 339/363 and 366/382 is repeated

Line 341

“..concentrate was obtained at 125 µm fraction”.

Means?:

concentrate was obtained at +250-125 µm fraction?

Line 342

97 % C. Fc       must be?    97% FC. FC

Line 342

“The Fc content of the final concentrate (C) in the finer fraction decreased continuously as 90.7% C for the +75-45 µm fraction and 82.6% C for – 20 µm fraction.”

What is the influence of entrainment? More intense phenomenon for fine particles.

You conducted several tests to define the influences of NaOH concentration, temperature and leaching time on preparation of high purity graphite.

What feed did you use? Graphite ore? Flotation concentrate? The use of flotation concentrate (high-grade feed) can decrease consumption and concentration of NaOH, or can also decrease temperature.

What was the liquid-solid ratio?

Line 386

Figure 13    must be    Figure 12

Figure 12 (a);  35% concentration of NaOH,    or  30% concentration of NaOH?

In materials and methods it is written 30%.

Figure 12 (a);  What was the temperature and time?

Figure 12 (b); What was the NaOH concentration time?

Figure 12 (c); What was the NaOH concentration temperature?

Line 442

“Where the treatment is done with pH was 8.5 and the chemical dosages were 187 g/t MIBC, 116 g/t kerosene 1500 g/t sodium silicate 204 g/t Flotanol 7026.”

Improved this paragraph

Line 450

The results showed that the graphite content of Käpysuo ore can be increased with alternative purifying methods like leaching chemicals (hydrochloric acid hydrofluoric acid) into  ultra-high-purity graphite (≥99.95% C) with fine particle sizes < 20 micron, which reaches the  requirement for using in a lithium-ion battery testing.

Improved this paragraph

The style of some references is not consistent.

Example:

[4] Chehreh Chelgani S; Rudolph M; Kratzsch R; Sandmann D; Gutzmer J. A review of graphite beneficiation techniques. Miner Process Extr Metall Rev 2016;37 (1):58-68.

Must be:

[4] Chehreh Chelgani, S.; Rudolph, M.; Kratzsch, R.; Sandmann, D.; Gutzmer, J. A review of graphite beneficiation techniques. Miner. Process. Extr. Metall. Rev. 2016, 37(1), 58-68.

Author Response

Response to Reviewer 4 Comments

Please see below, in red, our detailed response to comments.

Reviewer 4 Comments

The paper reports on a set of laboratory experiments for the study of “High-grade flake graphite deposits in metamorphic schist belt, Central Finland-Mineralogical and beneficiation of graphite in the lithium ion batteries application”

It is an interesting work, however it can only be accepted with minor changes.

I have listed some comments below

  • Page 1, Line 40, and ≥75 µm must be? and ≥45 µm. Done
  • Page 2 line 55, and –volcanic sequences     must be     and volcanic sequences. Done
  • Page 4, Table 1, N4442017R2 and N4442017R7; You did not show the weight of these two samples. What is the importance of showing the weight? We crossed the table 1
  • Was the N4442017_R7 sample the only one subjected to flotation tests? Four samples subjected to flotation tests, but we show in the manuscript only the N4442017_R7 sample result.
  • Fig 4, Do CT1, CT2, CT3, CT4 and CT5 products return to the previous cell? Yes.
  • Are the reagents, including frother, only added at the beginning of the process, before the two stages of rougher? The frother used in all stages as seen in Table 3
  • Line 245, screens; +10000 µm     must be     screens; +1000 µm: Done
  • Does Table 4 show the results for the size fractions?: +1400-1000? µm; +1000-710?; +710-500;   +45-20 and <20 µm. Yes
  • Does Figure 7 also show the results for the size fractions: +1400-1000? µm; +1000-710?; +710-500;   +45-20 and <20 µm. Yes
  • Does Figure 7 show the cumulative of the content of C? Figure 7a, b shown cumulative distribution function of particle sizes, while the carbon content in 7c, d not cumulative.
  • Are figures 7a and 7b correct? They represent the results (cumulative) of the content? (shown in Table 4) In Figure 7b, D80 is 100 µm and not 930 µm. Thanks for your note, we corrected the figure and the d values are d80= 850 and 930 µm, respectively).
  • Table 4 and Figure 7 show different results for the carbon content in the fractions +45-20 µm and <20 µm. For example, for the sample N4442017_R7, in table 4 those fractions present carbon contents of 10.5% and 19.56% and in Figure 7d the contents are 23.2% and 6.9%. Yes you’re right there is some print mistake from original table and we fixed that.
  • Page 10 Line 251; “It is evident from the histograms that the highest grades of graphitic carbon were obtained at size ranges larger than the 250 µm fraction in both samples. High carbon content occurs also in fine fraction +45-20 µm in sample N4442017_R7.” Updated
  • Is this correct? Or…..It is evident from the histograms that the highest grades of graphitic carbon were obtained at size fraction of +250/125 µm. Updated
  • But, based on table 4: High carbon content occurs also in fine fraction (<20 µm) for two samples. We updated the results of sieving and carbon content as seen in Table 4.
  • Has the effect of particle size on flotation been analysed? Yes, the d80 of the flotation feed samples was about 43 microns.
  • What is the effect of entrainment on the quality of the floated? The entrainment decreases the quality of the floated concentrate since very fine impurity particles are reported to the concentrate.
  • Table 3;  and line 336: What is Flotanol? Not described in the sub-chapter: material and methods; Flotanol C7 is alkyl polyglycole and explained that also in the Table 2 and in the text.
  • The use of only three wash flotation stages could lead to better results. Although the use of many (five) flotation stages increases the content of the float, it strongly penalizes recovery. Although stages 4 and 5 allow to increase the content, they caused a marked decrease in recovery. For example, for sample R7-3, for 5 stages the content is 84% and the recovery is 78.5%, but if you only used 3 stages you would get a content of 80.5% and a recovery of 97.5%: The objective of the study was to obtain a flotation graphite concentrate with C grade highest as possible. That is the reason for using 5 cleaner stages.
  • What is the influence of entrainment? More intense phenomenon for fine particles. Please see above
  • Page 13 line 327: “This product was the feed to flotation column, five stages of cleaner flotation…” In the cleaner flotation did you use a column or a cell? Its cell and was corrected in the text
  • (Line 330:     Table 9;       Line 386   Table 8) ; Corrected
  • Text between line 339/363 and 366/382 is repeated: Corrected
  • Line 341 concentrate was obtained at 125 µm fraction” Means?: concentrate was obtained at +250-125 µm fraction? Done
  • Line 342, 97 % C. Fc       must be?   97% FC. FC; Done
  • Line 342, “The Fc content of the final concentrate (C) in the finer fraction decreased continuously as 90.7% C for the +75-45 µm fraction and 82.6% C for – 20 µm fraction.” Corrected
  • You conducted several tests to define the influences of NaOH concentration, temperature and leaching time on preparation of high purity graphite. What feed did you use? Graphite ore? Flotation concentrate? The use of flotation concentrate (high-grade feed) can decrease consumption and concentration of NaOH, or can also decrease temperature. We used flotation concentrate
  • What was the liquid-solid ratio? Several purification test (LT1–LT21) were performed for graphite flotation concentrate. The effect of alkali concentration, roasting temperature and time was investigated (Table 7). Liquid–solid ratio was 2/1 (w/w) in alkaline roasting and 5/1 (w/w) in acidic leaching. The raw material was combined from flotation test concentrates CC5 and this material was used in all tests (please see the section; 2.4.2. Graphite Purification)
  • Line 386, Figure 13   must be   Figure 12; Corrected
  • Figure 12 (a); 35% concentration of NaOH, or 30% concentration of NaOH? Corrected
  • Figure 12 (a); What was the temperature and time? Done
  • Figure 12 (b); What was the NaOH concentration time? Done
  • Figure 12 (c); What was the NaOH concentration temperature? Done
  • Line 442 “Where the treatment is done with pH was 8.5 and the chemical dosages were 187 g/t MIBC, 116 g/t kerosene 1500 g/t sodium silicate 204 g/t Flotanol 7026.” Improved this paragraph: improved
  • Line 450; The results showed that the graphite content of Käpysuo ore can be increased with alternative purifying methods like leaching chemicals (hydrochloric acid hydrofluoric acid) into ultra-high-purity graphite (≥99.95% C) with fine particle sizes < 20 micron, which reaches the requirement for using in a lithium-ion battery testing. Improved this paragraph: Please see Table 7 graphite content of samples LT12 and LT14, fine particle sizes < 20 micron shown in SEM images Figure 10.
  • The style of some references is not consistent. Corrected
  • [4] Chehreh Chelgani S; Rudolph M; Kratzsch R; Sandmann D; Gutzmer J. A review of graphite beneficiation techniques. Miner Process Extr Metall Rev 2016, 37 (1):58-68.
  • Must be: [4] Chehreh Chelgani, S.; Rudolph, M.; Kratzsch, R.; Sandmann, D.; Gutzmer, J. A review of graphite beneficiation techniques. Miner. Process. Extr. Metall. Rev. 2016, 37(1), 58-68. Done

Round 2

Reviewer 2 Report

There are minor grammatical errors that still evident throughout the manuscript.

No additional comments.

Author Response

Dear Reviewer 2

We sincerely appreciate your valuable comments and we have been made grammatical errors accordingly.

Kindly Regards,

Reviewer 3 Report

The paper is still too long.  You have included too much unnecessary detail.  Please revise the paper to discuss the main findings in a clear, concise fashion.  It should not be much longer than 10 pages.

Author Response

Responses to Reviewer 3 Comments

The paper is still too long.  You have included too much unnecessary detail.  Please revise the paper to discuss the main findings in a clear, concise fashion.  It should not be much longer than 10 pages

Response: As requested, we have removed Figures 3, 5 and Tables 1, 4, 7  from the revised draft and we have submitted as Supplemental material figures and tables to reduce the pages of manuscript.